# Prompt Public Large Language Models to Synthesize Data for Private On-device Applications

**Shanshan Wu**,* **Zheng Xu**\*, **Yanxiang Zhang**\*, **Yuanbo Zhang, Daniel Ramage**
Google
{shanshanw, xuzheng, zhangyx, zyb, dramage}@google.com

## Abstract

Pre-training on public data is an effective method to improve the performance for federated learning (FL) with differential privacy (DP). This paper investigates how large language models (LLMs) trained on public data can improve the quality of pre-training data for the on-device language models trained with DP and FL. We carefully design LLM prompts to filter and transform existing public data, and generate new data to resemble the real user data distribution. The model pre-trained on our synthetic dataset achieves relative improvement of 19.0% and 22.8% in next word prediction accuracy compared to the baseline model pre-trained on a standard public dataset, when evaluated over the real user data in Gboard (Google Keyboard, a production mobile keyboard application). Furthermore, our method achieves evaluation accuracy better than or comparable to the baseline during the DP FL fine-tuning over millions of mobile devices, and our final model outperforms the baseline in production A/B testing. Our experiments demonstrate the strengths of LLMs in synthesizing data close to the private distribution even without accessing the private data, and also suggest future research directions to further reduce the distribution gap.

## 1 Introduction

While recent advances of machine learning models significantly benefit from scaling up both training data and model size (Kaplan et al., 2020; Anil et al., 2023; OpenAI, 2023a; Google, 2023; Touvron et al., 2023), smaller models have advantages in practical deployment due to inference latency, service cost, and privacy benefits when hosted on the local devices. User data are particularly effective to improve the performance of relatively small models targeting a specific task (Hard et al., 2018; Xu et al., 2023; Cho et al., 2024). Privacy-preserving methods are necessary for training these models using real user data (Bonawitz et al., 2022; Zhang et al., 2023b). Differential privacy (DP) (Dwork et al., 2006b; 2014), a mathematical guarantee applied to characterize the learning process, is a widely acknowledged method to prevent models from memorizing individual user's information in the training data. Cross-device federated learning (FL) (McMahan et al., 2017; Kairouz et al., 2021b), where devices collaboratively learn a model without transferring user data, is popular for limiting data access.

The usage of large-scale public and private data is important to achieve both privacy and utility for privacy-preserving methods. Training DP models from scratch to achieve state of the art utility with meaningful guarantees is challenging (Tramer & Boneh, 2020). Recent work (Li et al., 2021; Yu et al., 2021) show promising results with much better privacy utility trade-off by combining pre-training language models (LMs) on public data and fine-tuning on private data. Public pre-training has become a standard technique for DP training, FL (Nguyen et al., 2022), and the combination of DP and FL (Xu et al., 2023). In this paper, we focus on Gboard (Google Keyboard, a production mobile keyboard application), where small on-device LMs are trained and deployed (Xu et al., 2023). The training pipeline has two stages (see Figure 1): pre-training using server-side public data, and private fine-tuning

---

*Equal contribution.

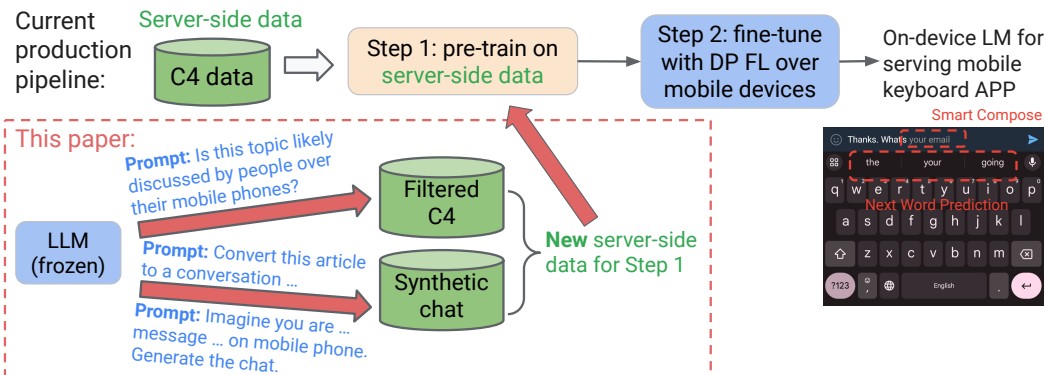

Figure 1: Overview of our experimental setup. It follows the two-step procedure of training on-device LMs for Gboard (Xu et al., 2023): 1) pre-training using server-side public data; followed by 2) fine-tuning on the private user data with DP FL. We use LLMs to synthesize data to replace the public C4 data (Raffel et al., 2020) in step 1.

over private user data with DP FL. The trained LM is deployed on the users' mobile devices to support features such as next word prediction, smart compose, smart completion and suggestion to improve the users' typing experience.

Pre-training on the server-side public data is particularly helpful in reducing the number of training rounds (and hence, the communication and computation cost) needed by DP FL over the private user data. Intuitively, pre-training allows a model to learn knowledge shared by the public and private domain, so that the privacy budget can be efficiently utilized during the private fine-tuning phase to learn important features specific to the private domain. Therefore, the closer the distribution between the public pre-training data and the private user data, the more savings in the privacy budget used by the DP FL. In this work, we explore whether the powerful large LMs (LLMs) can be used to improve the quality of the server-side pre-training data for Gboard.

LLMs with billions of parameters have achieved impressive performance in the general language generation and understanding tasks (see, e.g., (Anil et al., 2023; OpenAI, 2023a; Google, 2023; Touvron et al., 2023) and the references therein). As LLMs are a strong representation of their training data[1], Wang et al. (2023) asked *Can Public Large Language Models Help Private Cross-device Federated Learning*, and explored two approaches: 1) knowledge distillation (from the teacher LLM to student on-device LM) in pre-training, which significantly reduces the public data size and slightly improves the final performance after DP FL fine-tuning; and 2) distribution matching, which splits the privacy budget in two phases, and uses an LLM and a FL-trained LM from the first phase to filter the public data for the second phase. However, both approaches in (Wang et al., 2023) require non-trivial changes of the current public pre-training and DP FL fine-tuning pipeline. Moreover, Wang et al. (2023) did not fully exploit LLMs' emergent ability to generate long text sequences.

In this paper, we propose a simple yet effective method to improve the public data quality by exploiting the strong generative ability of LLMs. As shown in Figure 1, we carefully design the prompts to guide LLMs to generate data closer to the target domain. In our case, the target domain of Gboard is the private user typing data on their mobile phones. We investigate three types of LLM prompts: 1) filter and 2) transform the public C4 data (Raffel et al., 2020), and 3) generate diverse chat data by chain-of-thought (Wei et al., 2022) style prompting. The synthesized data can be directly used as the server-side pre-training data without extra changes to the DP FL phase, and hence, is simple to deploy in practice. Furthermore, the synthetic data can be potentially used as the proxy data for other tasks

---

[1]Pre-trained LLMs are considered to be public because their training data do not contain the on-device user data in Gboard. The privacy concerns of LLMs and their training data is an important independent topic (Brown et al., 2022; Tramèr et al., 2022).

(e.g., server-side evaluation) or models, which is another advantage over the previous methods proposed in (Wang et al., 2023).

The quality of our LLM generated data is assessed by running production FL experiments over the real user data from millions of mobile devices. Compared to the baseline C4 pre-training data (Xu et al., 2023), the LM pre-trained on our data gives relative improvement of 19.0% and 22.8% in the next word prediction accuracy when evaluated on the real user data (see Table 3). The LM also achieves superior performance in A/B testing after fine-tuning with DP FL (see Figure 4). Finally, we show that distribution gap between the public and private data can be further reduced, if a privately trained LM is available to filter the data.

## 2 Background

**Federated Learning (FL) and Differential Privacy (DP).** In cross-device FL, clients such as mobile devices collaboratively learn a model using the decentralized data. The Federated Averaging (FedAvg) algorithm and its variants (McMahan et al., 2017; Wang et al., 2021) are widely used in practice. In each training round (see Figure 2), the server first broadcasts a global model to a subset of clients; each client then updates their local model with local data, typically by an SGD optimizer, and sends back the model delta by subtracting the learned and initial local model weights; the model deltas are aggregated and used as pseudo gradient on the server to update the global model. After training typically thousands of rounds, the final model will be deployed on mobile devices for inference.

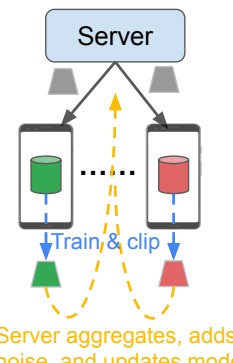

Figure 2: An FL training round with DP.

DP provides a quantifiable measurement of the privacy risk of models memorizing the individual user's information in the training data. Combining DP with FL gives an advanced privacy-preserving training method. DP is achieved by two operations (McMahan et al., 2018; Kairouz et al., 2021a; Choquette-Choo et al., 2024; Xu et al., 2023): 1) clipping the $l_2$ norm of each client's model delta to control their contribution, and 2) adding noise to the aggregated deltas on the server. In this paper, we use a production FL system similar to (Bonawitz et al., 2019) to run DP FL algorithm to train an on-device LM (see Section 4 for the setup). We fix the privacy and optimization parameters for fine-tuning with DP FL and only study the effectiveness of different pre-training public (proxy) data. See Appendix B for a mathematical description of the DP definition and details of the algorithms, and Appendix C for hyperparameters and DP guarantees.

**Synthetic Data Generation.** Using LLMs to generate synthetic data has shown promising results in many applications. Taori et al. (2023) used self-instruct (Wang et al., 2022) to fine-tune LLaMA 7B (Touvron et al., 2023) with synthetic instructions and answers generated by the large text-davinci-003 model (OpenAI, 2023b) with few-shot prompting. Eldan & Li (2023); Gunasekar et al. (2023); Li et al. (2023) used GPT-3.5 and GPT-4 models (Ouyang et al., 2022; Achiam et al., 2023) to generate data to train smaller models of fewer than 2 billion parameters for coherent storytelling, coding in Python, and common sense reasoning. Yu et al. (2024b) prompted LLMs with attributes to generate synthetic data similar to an existing attributed dataset. Zhu et al. (2023); Shu et al. (2023) used LLMs to filter and transform given text for the rewriting task. In this work, we use LLMs to synthesize data for Gboard, where the target domain distribution is user typing data on mobiles that are different from the public data on the web.

Private data can be used in various DP methods to guide LLMs to generate synthetic data close to the private distribution. These methods assume direct access to the private data for either directly fine-tuning an LLM (Kurakin et al., 2023; Yue et al., 2023; Yu et al., 2024a; 2023) or measuring the distance between the generated data and private distribution (Xie et al., 2024). It is challenging to apply these methods in a cross-device FL system due to the on-device resource limitations and privacy concerns of mobile users. Zhang et al. (2023a) proposed to prompt LLMs with classification labels to generate synthetic pre-training data

for FL. The method is designed for classification tasks and is studied for image and speech data, which cannot be directly applied to learn a language model.

Generating synthetic data to resemble the private domain in the FL setting is an active research area. Several concurrent papers (Hou et al., 2024; Li et al., 2024) explore data synthesis methods that require more interactions between the server and the private devices. This paper studies a simple prompt engineering approach, and provides unique value by examining the performance of synthesizing data in real-world production FL experiments.

## 3 Prompt LLMs to Synthesize Private-like Data

We design proper prompts to guide LLMs to process or generate data, so that the resulting data can be closer to the private target domain, compared to the baseline C4 dataset (Raffel et al., 2020) used by the current production (Xu et al., 2023). Our target domain distribution is formed by real-users' mobile keyboard typing data stored on their mobile devices. For privacy protection, we cannot directly collect or access the on-device examples. Instead, we use common sense knowledge to design the LLM prompts. While we focus on English, our approach can be easily applied to other languages. An instruction-tuned PaLM 2-S (Anil et al., 2023) is used as the LLM throughout the paper.

Three types of data are synthesized by the LLM: filtered C4, generated chat, and converted C4. While prompting LLMs to synthesize data has been explored previously as discussed in Section 2, to the best of our knowledge, we are the first to study this in a production FL application for private data, and validate its effectiveness by extensive experiments over millions of mobile phones.

### 3.1 Filter: Is the Public Example Likely Typed On Mobile Phones?

The public C4 dataset (Raffel et al., 2020) has over 360 million examples (782GB on disk), and is used as the pre-training data by the current production (Xu et al., 2023). Each example contains a paragraph of text extracted from a webpage. For each example, the LLM is prompted to output a binary answer:

*"Determine whether the following topic is likely to be discussed by people on their mobile phones. Give a score of 0 or 1, where 1 means very likely, and 0 means unlikely."*.

The filtered dataset has only 136GB on disk (around 30B tokens), about 17% of the original English C4 dataset, and is named *LLM-filter-C4-136G*. Table 1 lists a few positive and negative examples, showing that the LLM can understand the given example, and follow the instruction to make reasonable decisions for filtering.

| **In LLM-filter-C4-136G** |
| --- |
| "Beginners BBQ Class Taking Place in Missoula! Do you want to get better at ..." |
| "I thought I was going to finish the 3rd season of the Wire tonight ..." |
| "Weekend fun isn't just for humans. Dogs love to get out and enjoy some time away ..." |
| "One of the biggest games in the world is now available on smartphones ..." |
| **Not in LLM-filter-C4-136G** |
| "Discussion in 'Mac OS X Lion (10.7)' started by axboi87, Jan 20, 2012 ..." |
| "Foil plaid lycra and spandex shortall with metallic slinky insets. Attached metallic ..." |
| "How many backlinks per day for new site? Discussion in 'Black Hat SEO' started ..." |
| "BANGALORE CY JUNCTION SBC to GONDIA JUNCTION G train timings, routes ..." |

Table 1: Positive and negative C4 examples determined by the LLM over the query "whether the following topic is likely to be discussed by people on their mobile phones" (Section 3.1).

### 3.2 Prompt to Directly Generate Chat

We exploit the generative ability of LLMs to directly generate synthetic chat data. The key challenge is to ensure that the generated data are diverse (Gunasekar et al., 2023; Eldan & Li,

2023; Yu et al., 2024b). To improve diversity, we design the following prompts with seven variables:

*"Imagine you are a [GENDER] at age [AGE]. You are using the [CHAT-APP] APP to message [RECEIVER] on your mobile phone on the [TIME] of a [DAY]. You want to chat about the following topic: [TOPIC]. Generate the conversation between you and your message receiver. Do not include information other than the conversation."*.

Among the seven variables, five of them are sampled from a predefined set of categorical values, and two (RECEIVER and TOPIC) are self-generated by the LLM inspired by the chain-of-thought prompting (Wei et al., 2022). As shown in Figure 3, for given values of (AGE, GENDER, TIME, DAY, CHAT-APP), we first use LLM to generate a list of message receivers. Then we set RECEIVER to each value in the generated list, and use LLM again to generate a list of message topics. We post-process the generated list of receivers and topics to remove any duplications. Finally, we loop over the TOPIC in the generated list, and use LLM to generate the conversations.

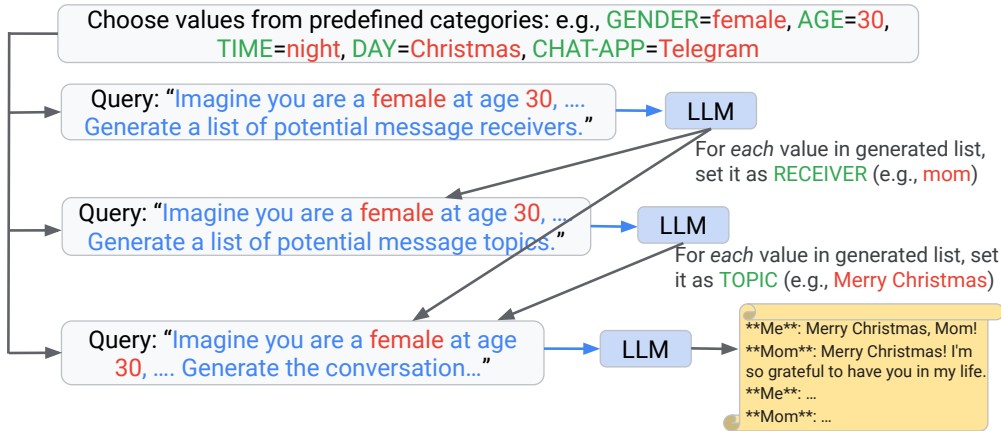

Figure 3: Overview of the procedure designed to increase the diversity of generated chat data. Given values for (AGE, GENDER, TIME, DAY, CHAT-APP), we sequentially use LLM to generate receivers, topics, and conversations (see details in Section 3.2).

We describe the predefined values of the five variables, and LLM prompts used to generate the other two variables (RECEIVER and TOPIC).

- AGE: Uniformly sampled from 15 to 55, and 3 age groups: "between 55 and 59", "between 60 and 64", and "over 65".

- GENDER: "male", and "female".

- TIME: "morning", "afternoon", and "night".

- DAY: 11 common holidays (e.g., "New Year's Day"), a special "vacation day", and 28 values in the format "[WEEKDAY] in the [SEASON]" where "WEEKDAY" can take 7 values from "Monday" to "Sunday", and "SEASON" can take 4 values from "spring" to "winter".

- CHAT-APP: "Android Messages", "Facebook Messenger", "Snapchat", "Instagram", "WhatsApp", "Discord", and "Telegram". As a sanity check of LLM's knowledge, we've asked the LLM to describe the differences between those popular chat apps, and verified that the answers are reasonable.

- RECEIVER: Given values for AGE, GENDER, TIME, DAY, and CHAT-APP, we ask the LLM to generate a list of message receivers: *"Imagine you are a [GENDER] at age [AGE]. You are using the [CHAT-APP] APP to message someone on your mobile phone on the [TIME] of a [DAY]. Generate a list of potential message receivers."*.

- TOPIC: Given values for AGE, GENDER, TIME, DAY, CHAT-APP, and a RECEIVER value generated by the LLM, we ask the LLM again to generate a list of topics:

> *"Imagine you are a [GENDER] at age [AGE]. You are using the [CHAT-APP] APP to message [RECEIVER] on your mobile phone on the [TIME] of a [DAY]. Generate a list of potential message topics."*.

We use top-k sampling (Fan et al., 2018) with $k = 40$ and temperature 0.2. A higher temperature usually gives a longer list of candidate receivers and topics. Because of the resource limitations, we choose a fixed temperature and leave the exploration of other sampling parameters or methods such as (Holtzman et al., 2020) as future work. The generated chat data has 19GB on disk (around 4B tokens), containing about 30 million multi-turn conversations (see Table 5 in the Appendix for a few examples).

### 3.3 Transform: Convert Public Example into Chat on Mobile Phones

Despite carefully designing the LLM prompts, the chat data generated by directly prompting the LLM (Section 3.2) can be less diverse than the C4 dataset. In Table 2, we compute the percentage of words in the vocabulary that have appeared in the dataset. The vocabulary contains 30K words and is used by the on-device LM over the en-US (United States) population. If a word does not appear in the pre-training data, then the corresponding word embedding will not be learned during the pre-training phase.

| Pre-training Data (suffix is data size) | % of en-US vocab covered |
|---|---|
| C4-782G (baseline) | 99.6% |
| LLM-filter-C4-136G | 99.6% |
| LLM-syn-chat-29G | 99.0% |
| - direct prompt | 79.3% |
| - convert C4 to chat | 99.0% |
| LLM-mix-166G | 99.9% |

Table 2: Percentage of words in the on-device LM vocabulary covered by pre-training data.

Therefore, a higher vocabulary coverage is usually desired. The synthetic chat data given by directly prompting the LLM has a lower vocabulary coverage 79.3% than the raw and filtered C4 datasets (both have 99.6%).

Motivated by this observation, we apply another approach to generate synthetic chat data: transform the LLM filtered C4 dataset (obtained in Section 3.1) into conversations. For each filtered C4 example, we ask the LLM to:

> *"Convert the following article to a conversation that you may message over your mobile phone. Generate the conversation. Include as many details as possible."*.

Due to the resource constraints, only 20% of the filtered C4 examples are converted to conversations. The resulting dataset has about 10GB size (around 2B tokens) and 10 million multi-turn conversations (see Table 6 in the Appendix for a few examples). As shown in Table 6, despite its small size, this dataset inherits a good vocabulary coverage 99.0% from the original C4.

### 3.4 Combine Filtered, Generated and Transformed Synthetic Data

We combine the 19GB data directly generated by LLM in Section 3.2 and the 10GB data transformed from C4 in Section 3.3, and obtain a chat dataset of 29GB with about 40 million multi-turn conversations. We name it *LLM-syn-chat-29G*. It exploits the generative ability of LLMs to synthesize chat to resemble the private user data in Gboard. Intuitively, the generated chat may have a distribution closer to the target distribution than the C4 data (see some evidence in Section 5). However, as we will show in Section 4.1, the LM trained on the LLM-syn-chat-29G alone achieves slightly lower accuracy than LLM-filter-C4-136G when evaluated on the real user data, possibly due to the diversity issue. Therefore, we combine LLM-syn-chat-29G and LLM-filter-C4-136G into *LLM-mix-166G*. We also tried different ratios when combining the two datasets (e.g., half from synthetic chat and half from filtered C4), and found that simply combining all the data works the best.

# 4 Experiments

As described in (Hard et al., 2018; Xu et al., 2023), our on-device LM is a one-layer LSTM (Hochreiter & Schmidhuber, 1997; Greff et al., 2016) with 670 hidden units, embedding dimension 96, and a 30K-size word-level vocabulary. The LM has about 6M parameters. As shown in Figure 1, we follow (Xu et al., 2023; Xu & Zhang, 2024) to train the LM in two steps: 1) server-side pre-training, and 2) fine-tuning with FL and DP. In DP FL fine-tuning, we run on a production FL system (Bonawitz et al., 2019) with the real user data. Experiments are performed over two populations of the mobile devices: United States (i.e., the "en-US" population) and India (i.e., the "en-IN" population).

To participate in a training round in cross-device FL system, the mobile devices have to satisfy local criteria such as being charging and connecting to unmetered network (Bonawitz et al., 2019; Huba et al., 2022), and minimum separation time across rounds (Xu et al., 2023). The total number of available devices are estimated to be around 13M for en-US and 8M for en-IN. Note that the exact population size is unknown as devices are not tracked or logged in the system.

## 4.1 Pre-training with Public Data on the Server

We use the standard cross entropy loss, and train the LM with Adam optimizer (Kingma & Ba, 2014) using 0.001 learning rate and $10^{-9}$ epsilon. The LM is trained for around 150K steps with batch size 20K. See Appendix C for more hyperparameter tuning details.

We evaluate the pre-trained LM on the decentralized user data in the en-US and en-IN populations. We use federated evaluation to aggregate metrics from multiple rounds of participated mobile devices. In each evaluation round, a subset of devices will receive the pre-trained LM and run evaluation on their data, and then the server aggregates the evaluation metrics from these devices. Minimum separation time criteria is enforced to guarantee different devices are chosen across rounds. The next word prediction (NWP) evaluation accuracy is reported in Table 3 with the following observations:

- Pre-training on the synthetic chat data LLM-syn-chat-29G gives a lower accuracy than pre-training on the filtered C4 "LLM-filter-C4-136G", potentially because that the synthetic chat has smaller size and lower diversity as discussed in Section 3.3.

- Combining the filtered C4 and the synthetic chat data gives the best pre-training dataset "LLM-mix-166G", which achieves 22.8% and 19.0% relative improvement over the baseline C4 data for the en-US and en-IN populations, respectively.

**Discussion on the quality measure.** Evaluating the quality of the generated data is a common challenge. It is even more challenging when the target domain is the private user data, because we need to be extremely careful on privacy protection. In this work, we measure the data quality by training an on-device LM and using federated evaluation to aggregate a single scalar value of NWP accuracy. Developing practical privacy-preserving methods to measure the distribution similarity between the server-side data and the private on-device data is an important future work. A related work is (Hou et al., 2023), which proposes a method FreD to measure the distance between a server-side dataset and the private on-device data, and use it to select the best server-side dataset for pre-training. Applying FreD in practice and combining with our synthetic data approach is an interesting direction for future work. Note that this requires careful allocation of privacy budgets in FreD and federated fine-tuning.

## 4.2 Fine-tuning with Differentially Private Federated Learning

After pre-training on the server in Section 4.1, we follow the recommended practices in (Xu et al., 2023) to fine-tune the LM using a production FL system (Bonawitz et al., 2019). For more details about the federated training algorithm and DP accounting, see Section 2 and Appendix B.

| Pre-training Data (suffix is data size) | NWP Accuracy on user data (en-US) | NWP Accuracy on user data (en-IN) |
|---|---|---|
| C4-782G (baseline) | $0.1182 \pm 0.0002$ | $0.0441 \pm 0.0004$ |
| LLM-filter-C4-136G | $0.1425 \pm 0.0004$ | $0.0491 \pm 0.0005$ |
| LLM-syn-chat-29G | $0.1308 \pm 0.0002$ | $0.0401 \pm 0.0004$ |
| **LLM-mix-166G** | **$0.1452 \pm 0.0003$ (+22.8%)** | **$0.0525 \pm 0.0005$ (+19.0%)** |

Table 3: Next word prediction (NWP) accuracy evaluated on user data in the United States (en-US) and India (en-IN) population. The LMs are trained on the server-side data as described in Section 4.1, i.e., only pre-training and no fine-tuning on user data. For each LM, we run federated evaluation at three different times during a week (each run lasts for a day collecting metrics from more than 100K devices). Mean and standard deviation over the three federated evaluation runs are reported. LMs pre-trained on LLM-mix-166G data achieves the best accuracy with 22.8% and 19.0% relative improvement over the baseline.

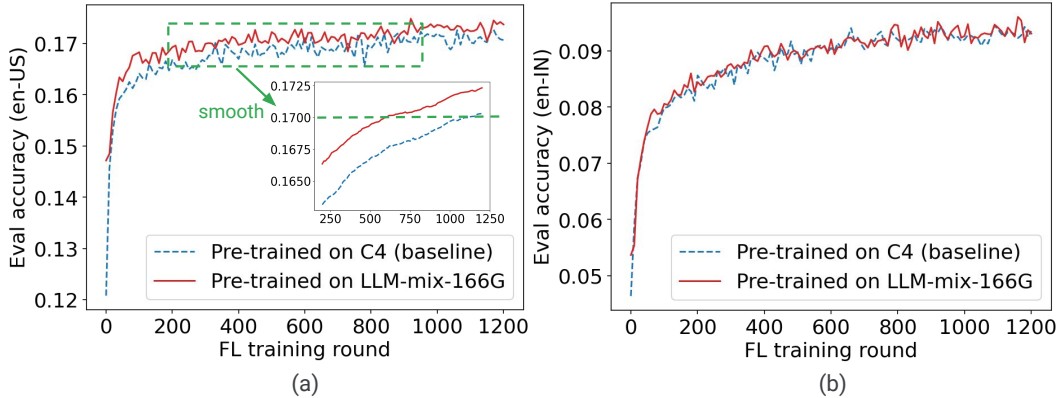

Figure 4: NWP evaluation accuracy for fine-tuning models with DP FL over the real mobile devices in the (a) United States and (b) India populations. Compared to the baseline of pre-training on C4, the LM pre-trained on LLM synthetic data achieves higher initial accuracy, and also maintains superior or comparable accuracy during the fine-tuning process.

As the FL training proceeds, we run federated evaluation of the trained LMs over a different set of devices (i.e., the holdout set) in the same population, and report the NWP evaluation metrics in Figure 4. We highlight the following observations:

- Compared to the baseline (C4 pre-trained LM), the LM pre-trained on LLM data has a higher accuracy at training round 0. This is consistent with the metrics reported in Table 3, which are potentially aggregated from more devices across multiple federated evaluation rounds.

- During the FL training, the LM pre-trained on the LLM data maintains superior (over the en-US population) or comparable evaluation accuracy (over the en-IN population). Specifically, to reach the 0.17 accuracy on the en-US population, our method needs around 600 rounds while the the baseline needs around 1100 rounds (i.e., almost 2x more), giving a significant saving in the communication and computation cost and an improvement in the privacy guarantees.

**A/B testing.** After fine-tuned with DP FL, we conduct live A/B testing to measure the LM performance in the production environment. Specifically, we measure two metrics: WMR (Word Modified Ratio, i.e., the ratio of words being modified during typing or after committed) and WPM (Word Per Minute, i.e., the number of committed words per minute). For fairness, the LMs in A/B testing have the same privacy guarantees, achieved by using the same noise multiplier, same number of clients per round, same minimum separation time, and same number of training rounds (see Appendix C.3 for more details). For en-US, the FL fine-tuned English LM pre-trained on LLM data improves over the baseline WMR by

0.64% and WPM by 0.11%. For en-IN, the FL fine-tuned English LM pre-trained on LLM data improves over the baseline WMR by 0.05% and WPM by 0.04%. These improvements, especially in en-US, are significant for improving the users' mobile typing experience.

**Discussion on en-US vs en-IN.** Our LLM synthetic data is more effective in the en-US population than the en-IN population. This indicates that the synthetic data may be less similar to the real user typing data in India. An interesting direction of future work is to see if using target country/region in the LLM prompts can help synthesize better data.

## 5   Improve Synthetic Data with Fine-tuned On-device LM

| Data (suffix is size) | NWP accuracy | LLM-prox-32G data size | |
|---|---|---|---|
| C4-782G (baseline) | $0.1182 \pm 0.0002$ | Total | 32GB (19% LLM-mix-166G) |
| LLM-mix-166G | $0.1452 \pm 0.0003$ | - Filter C4 | 17GB (12% LLM-filter-C4-136G) |
| LLM-prox-32G | **0.1509** $\pm$ 0.0004 | - Syn chat | 15GB (52% LLM-syn-chat-29G) |

Table 4: **Left** table compares the NWP accuracy evaluated over the en-US population for LMs trained on different datasets. We follow the same federated evaluation process as in Table 3. The filtered LLM data achieve higher accuracy compared to the unfiltered LLM-mix-166G, despite having a much smaller size of 32GB as shown in the **Right** table. Results on the en-IN population can be found at Appendix D.

So far we only use the common sense knowledge to prompt the LLM to synthesize data closer to the private distribution of user typing data in Gboard. While the LLM-based synthetic data provide impressive gains compared to the C4 baseline in pre-training as shown in Table 3, the on-device LM still benefits a lot from fine-tuning over the real user data as shown in Figure 4. This indicates a gap between the distribution of the LLM-based synthetic data and that of the real user data. In this section, we present a preliminary study of a simple strategy to close the gap, without changing the privacy-preserving way of accessing private data. Specifically, we ask the following question: *can we reduce the distribution gap by using a fine-tuned on-device LM from Section 4.2 to filter the data?*

**Note that the goal of this preliminary study** is *not* training new on-device LMs as in Section 4.2, instead, the goal is to see if we can use an *existing* privately-trained LM to obtain a better server-side dataset. This new dataset can be of independent interest for server-side tasks such as research simulation in the datacenter, and improving server-side models that cannot be easily fine-tuned with DP FL. The process of obtaining this new server-side dataset (i.e., using an *existing* privately-trained LM to filter the original data) does not incur *extra* privacy cost, following the post-processing property of differential privacy. We did not use this new dataset to further pre-train an on-device LM, and fine-tune it with DP FL as in Section 4.2. We can potentially adopt the mid-training approach in (Wang et al., 2023), which will complicate the standard production pipeline and need specialized privacy accounting.

For each example in the LLM-mix-166G dataset, we compute 3 values: 1) OOV rate: the percentage of tokens not in the LM vocabulary; 2) pre-trained LM score: the average log-likelihood computed by the LM trained on LLM-mix-166G as in Section 4.1; 3) fine-tuned LM score: the average log-likelihood computed by the LM pre-trained on LLM-mix-166G and fine-tuned over real user data with DP FL as in Section 4.2. Examples satisfying the following conditions are kept in the filtered LLM data (see Appendix C for more details): OOV rate $\leq 0.6$; fine-tuned LM score $\geq 5$; fine-tuned LM score $\geq$ pre-trained LM score.

After filtering LLM-mix-166G, the data size is decreased to 32GB (around 7B tokens). We use *LLM-prox-32G* to represent this new proxy dataset. We follow the same step described in Section 4.1 to measure the quality of LLM-prox-32G: train an LM from scratch on the data, and then perform federated evaluation over the real user data. As shown in Table 4, on the en-US population, LLM-prox-32G further improves the accuracy from 0.1452 to 0.1509 compared to LLM-mix-166G, and achieves much higher accuracy than the baseline C4. This indicates that the filtered data have higher quality, despite only having 19% of the original

LLM-mix-166G data. Table 4 also shows that a larger fraction of LLM filtered C4 examples are filtered out compared to the synthetic chat data, which potentially indicates that the majority of the LLM filtered C4 are less similar to the private user data.

## 6 Conclusion

Inspired by recent advances in generative LLMs, this work studies how LLMs can help differentially private federated learning for training small on-device models. In particular, we focus on Gboard (Google Keyboard, a production mobile keyboard application) where the task is to learn a small on-device LM using the private user typing data. Our work provides evidence that even with no access to the private data, common sense knowledge and careful prompt design can help guide LLMs to synthesize data similar to the target domain. Effectiveness of our method is verified by extensive FL experiments over the real-world user data from the millions of mobile devices.

Our results suggest a few interesting directions of future work, including developing more privacy-preserving quality measures, investigating other LLM prompting strategies such as adding the country/region information, and different sampling methods, as already pointed out in Section 4.1, Section 4.2, and Section 3. Moreover, given the promising results from the preliminary study in Section 5, further investigation on improving the quality of proxy data with guaranteed privacy-preserving methods can be a rewarding direction.

**Acknowledgments**

The authors would like to thank H. Brendan McMahan, Sewoong Oh, and Da Yu for early discussions on this work; Zachary Garrett, Hui Li, Michael Riley, Jesse Rosenstock, and Shumin Zhai for valuable comments on an early draft; Jeremy Gillula and Jen Wei for internal review process.

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

| **Example 1** |
| --- |
| **LLM query:** |
| Imagine you are a female at age 23. You are using the Android Messages APP to message your family on your mobile phone on the afternoon of a vacation day. You want to chat about the following topic: I can't wait to come home and tell you all about it. Generate the conversation between you and your message receiver. Do not include information other than the conversation. |
| **LLM output:** |
| `**Me:**` Hey mom, I'm having so much fun on vacation! I can't wait to come home and tell you all about it. `**Mom:**` That's great to hear! I'm so glad you're enjoying yourself. `**Me:**` I am! I've been swimming, sunbathing, and exploring the island. I've also made some new friends. `**Mom:**` That sounds like a lot of fun! I'm sure you'll have some great stories to tell us when you get home. `**Me:**` I know I will! I'm also looking forward to seeing you and dad again. `**Mom:**` We're looking forward to seeing you too! Have a safe trip home. `**Me:**` I will. Love you! `**Mom:**` Love you too! |
| **Example 2** |
| **LLM query:** |
| Imagine you are a male at age 50. You are using the WhatsApp APP to message your therapist on your mobile phone on the afternoon of a Monday in the autumn. You want to chat about the following topic: I'm feeling overwhelmed and don't know how to cope. Generate the conversation between you and your message receiver. Do not include information other than the conversation. |
| **LLM output:** |
| `Therapist:` Hi [client name], how are you doing today? `Client:` I'm not doing so well. I'm feeling overwhelmed and I don't know how to cope. `Therapist:` What's been going on that's making you feel overwhelmed? `Client:` Well, a lot of things. I'm having trouble at work, my relationship is on the rocks, and I'm just not feeling like I'm good enough. `Therapist:` It sounds like you're going through a lot right now. It's understandable that you're feeling overwhelmed. `Client:` I just don't know what to do. I feel like I'm drowning. `Therapist:` It's okay to feel overwhelmed. It's not easy to deal with all of this at once. `Client:` I know, but I don't know how to make it stop. `Therapist:` There are a few things you can do to help cope with feeling overwhelmed. One thing is to talk to someone you trust, like a friend, family member, or therapist. Talking about what you're going through can help you feel less alone and can give you some perspective. Another thing you can do is to take some time for yourself each day. This could mean taking a walk, reading a book, or spending time in nature. Taking some time to relax and de-stress can help you feel more in control. Finally, it's important to remember that you're not alone. There are people who care about you and want to help. If you're feeling overwhelmed, reach out for help.There are resources available to help you cope with stress and anxiety. `Client:` Thank you. I'll try to do those things. `Therapist:` You're welcome. I'm here for you if you need anything. |

Table 5: Example conversations generated by directly prompting the LLM described in Section 3.2. Note the both the receiver and topic shown in the LLM query above are also generated by the LLM in separate queries.

# A   Synthetic Chat Examples

Table 5 and Table 6 show examples of synthetic chat by direct generation (Section 3.2) and converting C4 (Section 3.3), respectively.

| Example 1 |
|---|
| **LLM query:** |
| Convert the following article to a conversation that you may message over your mobile phone. Generate the conversation. Include as many details as possible. Our hobby is our dogs (and Ragdoll cats). They are first and foremost our companions and we love to train with them Obedience and Agility. If everything regarding health and temperament checks out, we breed them after carefully selecting an appropriate mate. The puppies are raised in our living room and socialized to all kinds of regular household sights and sounds. |
| **LLM output:** |
| `**Me:**` Hey, I saw your ad for the puppies for sale. I'm interested in learning more about them. `**You:**` Great! Our dogs are our hobby and we love to train them in obedience and agility. We only breed them after carefully selecting an appropriate mate and raising the puppies in our living room so they're socialized to all kinds of regular household sights and sounds. `**Me:**` That sounds great! What kind of dogs do you have? `**You:**` We have a variety of breeds, including Golden Retrievers, Labrador Retrievers, and Poodles. `**Me:**` I'm interested in a Golden Retriever. Do you have any available? `**You:**` Yes, we have a litter of Golden Retriever puppies that will be ready to go home in a few weeks. `**Me:**` Can I come by to see them? `**You:**` Sure, just let me know when you're available. `**Me:**` I'm free tomorrow afternoon. `**You:**` Great, I'll see you then. `**Me:**` Looking forward to it! |
| Example 2 |
| **LLM query:** |
| Convert the following article to a conversation that you may message over your mobile phone. Generate the conversation. Include as many details as possible. Five teachers from the Netherlands came over this week to see our school and how our education system works in England. They were so impressed with our children and the way we do things here at Ramridge. It was interesting to learn about their school system too! |
| **LLM output:** |
| `**Me:**` Hey, did you hear about the five teachers from the Netherlands who came to visit our school this week? `**Friend:**` No, I didn't! What were they here for? `**Me:**` They wanted to see our school and how our education system works in England. They were really impressed with our children and the way we do things here at Ramridge. `**Friend:**` That's great! I'm glad they had a good experience. `**Me:**` Me too. It was also interesting to learn about their school system. It's a lot different from ours. `**Friend:**` In what ways? `**Me:**` Well, for one thing, they start school at a much younger age. And they have a lot more emphasis on STEM subjects. `**Friend:**` That's interesting. I wonder if it's more effective than our system. `**Me:**` I don't know. It's hard to say. But it's definitely worth looking into. `**Friend:**` Yeah, I agree. It sounds like they're doing some things right. `**Me:**` Definitely. I'm glad we had the opportunity to learn from them. `**Friend:**` Me too. It was a great experience. |

Table 6: Example conversations generated by asking LLM to convert filtered C4 examples into chats (see Section 3.3 for more details).

# B   Formal DP Definition and DP FL Algorithm

DP provides a quantifiable measurement of the privacy risk of models memorizing the individual user's information in training data. We provide a formal definition of $(\epsilon, \delta) - DP$ (Dwork et al., 2006a; 2014),

**Definition B.1** $((\epsilon, \delta)$-Differential Privacy)**.** A randomized algorithm $\mathcal{M}$ satisfies $(\epsilon, \delta)$-DP for $\mathbb{D}$ if for any two neighboring datasets $\mathbb{D}, \mathbb{D}'$ and for all $\mathcal{S} \subset \text{Range}(\mathcal{M})$:

$$\Pr[\mathcal{M}(\mathbb{D}) \in \mathcal{S}] \leq e^{\epsilon} \Pr[\mathcal{M}(\mathbb{D}') \in \mathcal{S}] + \delta.$$

DP can be combined with FL for advanced privacy-preserving training method, where the neighboring datasets $\mathbb{D}, \mathbb{D}'$ are defined by changing all the data of a user device.

The Federated Averaging (FedAvg) algorithm and its variants (McMahan et al., 2017; Wang et al., 2021) are widely used in practical FL systems. In training round $t$ of generalized FedAvg, the server first broadcasts a global model $w^t$ to a subset of clients; each client $i$ then updates their local model $\theta$ with their local data, typically by a SGD optimizer, and sends back the model delta between learned and initial local model weights $\Delta_i^t = \theta_i^t - w^t$;the model deltas are aggregated $\Delta^t = \sum_i \Delta_i^t$ and used as pseudo gradient on the server to update the global model. After training typically thousands of rounds, a final model can be deployed on mobile devices. DP is achieved by clipping the $l_2$ norm of the model delta to control the contribution of each client, and then adding noise to the aggregated deltas on the server. The recent DP-Follw The Regularized Leader (DP-FTRL) (Kairouz et al., 2021a; Choquette-Choo et al., 2024) algorithms add correlated noise across rounds to achieve strong privacy and utility at the same time, which are used in training production on-device LMs in cross-device FL (Xu et al., 2023). We use the tree aggregation DP-FTRL (Kairouz et al., 2021a) in the generalized FedAvg following (Xu et al., 2023), as detailed in algorithm 1.

---

**Algorithm 1** FedAvg (McMahan et al., 2018) with DP-FTRL (Kairouz et al., 2021a) for DP FL

**input** : clients per round $m$, learning rate on client $\eta_c$ and on server $\eta_s$, momentum $\beta = 0.9$, total number of rounds $T$, noise multiplier for model delta $z$, clip norm $C$

Initialize model $w^0$ with pretraining
Initialize momentum buffer $\bar{P}^0 = 0$
Initialize DP tree state $\mathcal{T}$ with $zC$
**for** each round $t = 1, 2, \dots, T$ **do**
  $\mathcal{Q}^t \leftarrow$ (at least $m$ users for this round)
  **for** each user $i \in \mathcal{Q}^t$ **in parallel do**
    $\Delta_i^t \leftarrow$ ClientUpdate$(i, w^{t-1})$
  $P^t, \mathcal{T} \leftarrow$ PrivateSum$(P^{t-1}, \mathcal{T}, \frac{1}{m}\sum_{i\in\mathcal{Q}^t}\Delta_i^t)$
  $\bar{P}^t = \beta\bar{P}^{t-1} + P^t, w^t \leftarrow w^0 + \eta_s\bar{P}^t$

**function** ClientUpdate$(i, \theta)$
  $\mathcal{G} \leftarrow$ (user $i$'s local data split into batches)
  **for** batch $g \in \mathcal{G}$ **do**
    $\theta \leftarrow \theta - \eta_c \nabla \ell(\theta; g)$
  $\Delta \leftarrow \theta - \theta_0$
  $\Delta' \leftarrow \Delta \cdot \min\left(1, \frac{C}{||\Delta||}\right)$
  **return** $\Delta'$

---

# C   Hyperparameter Tuning and DP Guarantees

## C.1   Hyperparameters for pre-training (Section 4.1)

The LM is trained in an auto-regressive manner. Before pre-training, we pre-process the datasets and treat each turn in multi-turn chats as a single training example, and each sentence in the filtered C4 data as a single training example. The learning rates are tuned among $\{0.0001, 0.001, 0.01\}$ and epsilons are tuned among $\{10^{-7}, 10^{-8}, 10^{-9}\}$. For all the pre-training datasets, 150K steps are enough for the training to converge. Pre-training for fewer (e.g., 100K) or more (e.g., 200K) steps did not help the evaluation accuracy over the real user data.

## C.2 Hyperparameters for filtering LLM data (Section 5)

The fine-tuned LM score was tuned over three thresholds {6, 5, 4}. We also compared with and without the constraint of fine-tuned LM score ≥ pre-trained LM score, and found that adding the constraint slightly helps. The OOV rate was chosen based on a manual inspection over the examples with very high fine-tuned LM scores and very low pre-trained LM scores (these examples usually have low quality and high OOV rate).

## C.3 Reporting privacy guarantees

This section clarifies the nuances of the DP guarantees following the guidelines outlined in (Ponomareva et al., 2023, Sec. 5.3). The main differences compared to (Xu et al., 2023) is the usage of the public data in pre-training, and the exact DP guarantees for model training. We include the entire checklist for completeness.

1. **DP setting**. This is a central DP guarantee where the service provider is trusted to correctly implement the mechanism.

2. **Instantiating the DP Definition**

   (a) *Data accesses covered*: The DP guarantee applies to all well-behaved clients[2] in a single training run. We do not account for hyperparameter tuning, or the selection of the final model checkpoint using evaluation metrics or A/B testing in our guarantees. Public data such as C4 (Raffel et al., 2020; Xue et al., 2020) or LLM-based synthetic data, are used for pre-training.

   (b) *Final mechanism output*: Only the final model checkpoint is released for production deployment, however the mechanism's output is technically the full sequence of privatized gradients, and so the guarantee also applies at this level, and hence all intermediate models are protected (including those sent to devices participating in federated learning).

   (c) *Unit of privacy*. Device-level DP is considered, i.e., the notion of adjacency is with respect to arbitrary training datasets on each client device, and the device might have an arbitrarily large local dataset containing arbitrary training examples. For user's with a single device, this corresponds directly to user-level DP; for devices shared with multiple users, this provides a stronger notion of DP than user-level; for a user with multiple devices that happen to both participate in training the model, the notion is weaker, but group privacy can be used to obtain a user-level guarantee.

   (d) *Adjacency definition for "neigbouring" datasets*: We use the zero-out definition (Kairouz et al., 2021a). This is a a special form of the add-or-remove definition, where neighboring data sets differ by addition/removal of a single client. In the absence of a client at any training step, we assume that the client's model update gets replaced with the all zeros vector. This assumption enforces a subtle modification to the traditional definition of the add/remove notion of DP which allows neighboring data sets to have the same number of records.

3. **Privacy accounting details**

   (a) *Type of accounting used*: Both $\rho-$zCDP (Bun & Steinke, 2016) accounting, and PLD accounting (DP Team, 2022) for $(\epsilon, \delta)-$DP are used.

   (b) *Accounting assumptions*: Each client only participates limited times during the training, and there are at least a min-separation number of rounds between two consecutive participation of a client. Client participation is enforced by a timer on clients in the cross-device FL system.

   (c) *The formal DP statement*: The en-IN Gboard LM with 1200 rounds has $0.42-$zCDP, which can be transformed to $(\epsilon = 5.95, \delta = 10^{-10})-$DP using

---

[2]Clients that faithfully follow the algorithm including participation limits. Due to the design of the algorithm, a mis-behaved client does not adversely affect the DP guarantee of any well-behaved clients.

PLD accounting (DP Team, 2022). This is improved from (Xu et al., 2023) because a larger min separation is achieved. The en-US Gboard LM with 1200 rounds has $0.5-$zCDP, and alternatively, $(\epsilon = 6.55, \delta = 10^{-10})-$DP

(d) *Transparency and verifiability*: We use the open-sourced core implementation code in TensorFlow Federated and Tensorflow Privacy. Key portions of the cross-device FL system are also open sourced.

# D   More Experimental Results

We repeat the same experimental procedure in Section 5 for the en-IN population, and report the results in Table 7. Compared to the en-US results in Table 4, on the en-IN population, filtering the LLM data by FL-trained LM does not seem to help reduce the distribution gap between the LLM data and the real user data. This is possibly because that the LLM-mix-166G overlaps less with en-IN data distribution, as discussed in Section 4.2.

| Data | NWP accuracy | | Filtered LLM data size | |
|------|--------------|---|------------------------|---|
| C4-782G (baseline) | $0.0441 \pm 0.0004$ | Total | 32GB (19% LLM-mix-166G) | |
| LLM-mix-166G | $\mathbf{0.0525} \pm 0.0005$ | - Filter C4 | 18GB (13% LLM-filter-C4-136G) | |
| Filtered LLM data | $\mathbf{0.0518} \pm 0.0004$ | - Syn chat | 14GB (48% LLM-syn-chat-29G) | |

Table 7: **Left** table compares the NWP accuracy evaluated over the en-IN population for LMs trained on different datasets. We follow the same federated evaluation process as in Table 3. The filtered LLM data achieve similar accuracy compared to the unfiltered "LLM-mix-166G", while having a much smaller size (32GB vs 166GB as shown in the **Right** table).

