# OpenReview forum: "Prompt Public Large Language Models to Synthesize Data for Private On-device Applications"
_colmweb.org/COLM/2024/Conference — COLM_

### Official Review · Reviewer_X9QH · 2024-04-30

**Rating:** 6
**Confidence:** 3
**Ethics Flag:** 1

**Summary:**

This work uses Large Language Models (LLMs) to filter and generate real/synthetic data for Data Privacy (DP) and Federated Learning (FL) purposes. A model pre-trained on the final dataset generated by this work is able to improve next-word prediction (NWP) accuracy against a strong baseline. This work has been tested and evaluated on real large-scale data across different devices and can be used as a reference for industry researchers (the federated experiments at that level could be prohibitive for academics). The experiments also highlight aspects such as vocabulary coverage and its impact on the final accuracy results.
NWP Accuracy is improved (+19%) using a dataset that is significantly smaller (166GB vs 782GB) showing the value of data filtering, data generation, and data processing/formatting.

**Reasons To Accept:**

This paper conducts experiments at a large scale over different locales (US, India) and highlights the strengths of LLM-generated data/summaries. The experiments also show how experiments translate across locales and potential weaknesses when improvements over one locale do not translate to a different region.

**Reasons To Reject:**

The ablation studies and experiments show interesting results, but the main techniques/methods have been published already. The large-scale experimentation adds some novelty, but the novel contributions might not be enough. Also, the NWP metric should probably be used along some perplexity scores for a larger/clearer picture.

---

> ### Author Rebuttal · Authors · 2024-05-31
>
> Thanks for your time and valuable feedback!
>
> > "the NWP metric should probably be used along some perplexity scores..."
>
> We ran extra experiments to collect the perplexity metrics for the pre-trained LMs in en-US:
> |Pre-training Data|NWP|Perplexity|
> |-------|-----|-------|
> |C4-784G (baseline)|0.1176|813.4|
> |LLM-filter-C4-136G|0.1415|526.4|
> |LLM-syn-chat-29G |0.1295| 1294.7 |
> |LLM-mix-166G |0.1440|508.5|
>
> Note:
>
> - For fair comparison, the numbers were collected this week (during May 27-29) over 100M+ examples from 200k+ mobile devices. That's why the NWP values are slightly different from Table 3 of our paper. Note that this week overlaps with Memorial Day, which may affect the result.
>
> - While the perplexity scores seem high, they are in the normal range from our previous experience. LLM-mix-166G achieves both the best NWP and perplexity. The perplexity generally aligns with NWP accuracy, though not always, which is also normal from our previous experience. In this paper, we follow [4] and focus on NWP, because it is shown to **correlate well** with the live metrics in A/B testing (which we also report in Section 4). Besides, reporting NWP accuracy is standard for federated learning LM experiments [1,2,3].
>
> [1] Reddi et al., “Adaptive Federated Optimization”, ICLR 2021
>
> [2] Nguyen et al., “Where to Begin? On the Impact of Pre-Training and Initialization in Federated Learning”, ICLR 2023
>
> [3] Li et al., “Private Adaptive Optimization with Side information”, ICML 2022
>
> [4] Xu et al., “Federated learning of gboard language models with differential privacy”, ACL industrial track, 2023
>
> > "...the novel contributions might not be enough."
>
> Though prompting LLMs to synthesize data has been explored before, we specifically designed the prompts to mimic user data in mobile virtual keyboards. Besides, we are the first to systematically study this in a production federated learning application and verify it over real-user data from millions of mobile devices. Running large-scale real-world experiments is challenging because of complex system issues and user behaviors.
>
> There are a few interesting and inspiring observations from our production experiments: e.g., LLM synthetic data are closer to the private distribution but there is still a gap (Figure 4); LLM filtered C4 is better than LLM generated chat (Table 3); using a privately-trained LM to filter data only gives a slight gain (Table 4). We hope that our work can inspire more future research in this area.

---

> > ### Comment · Reviewer_X9QH · 2024-06-04
> >
> > Thank you for the answers.

---

### Official Review · Reviewer_NTtL · 2024-05-09

**Rating:** 6
**Confidence:** 4
**Ethics Flag:** 1

**Summary:**

This paper investigates how large language models (LLMs) trained on public data can improve the quality of pre-training data for the on-device language models trained with DP and FL. The authors design LLM prompts to filter and transform existing public data and generate new data that resembles the distribution of real user data. Compared to baseline models pre-trained on standard public datasets, models pre-trained on their composite datasets achieved relative improvements in next word prediction accuracy of 19.0% and 22.8% when evaluating real user data. It clearly outlines the problem addressed, the approach taken, and the outcomes.

**Questions To Authors:**

1. Whether the effect of filtering and merging data by modifying the prompt can also be achieved by other data enhancement methods. This is not compared in the paper.
2.The two methods mentioned in the paper (Wang et al., 2023) are not compared in the experiment.
3.The vocab coverage of convert C4 to chat and LLM-SYN-Chate-29G is the same. Is the effect of direct mix filter-C4 and convert C4 to chat much worse than existing mix methods? If there is no big difference in effect, this seems to save a lot of computational overhead.

**Reasons To Accept:**

This paper proposes a simple yet effective method to improve the public data quality by exploiting the strong generative ability of LLMs. It provides an effective solution for how to better process data sets with the generation power of LLMs to better adapt to FL and DP.

**Reasons To Reject:**

1. The selected baseline methods are not representative.
2. Lack of ablation experiments.

---

> ### Author Rebuttal · Authors · 2024-05-31
>
> Thanks for your time and valuable feedback!
>
> > 1. "Whether ... be achieved by other data enhancement methods."
>
> The space of the LLM prompts (and other data methods) is too large to be explored in a real-world production system. Our key contribution is to show that even a simple idea can work. We are the first to verify this simple idea in a production federated learning application over real-user data from millions of mobile devices. Running large-scale experiments is challenging due to complex system issues and real-user behaviors. We hope that our work can inspire more future research.
>
> > 2. "...the paper (Wang et al., 2023) are not compared in the experiment."
>
> We did not compare it, for two reasons:
>
> - As pointed out in the Introduction, the two approaches “require non-trivial changes to the current" production pipeline. E.g., knowledge distillation requires changing the on-device LM’s original (word-level) tokenizer to LLM's wordpiece tokenizer, which is difficult from production deployment standpoint. Distribution matching, an enhancement on top of knowledge distillation, adds another layer of complexity (requires careful allocation of privacy budgets).
>
> - Besides, Wang et al., 2023 did not explore the LLM generative abilities, which is the focus of our paper.
>
> > 2. "...Is the effect of direct mix filter-C4 and convert C4 to chat much worse...?"
>
> Following your advice, we created a new dataset by merging LLM-filtered-C4-136g and LLM-converted-C4-10g, and evaluated the pre-trained LMs:
>
> | Pre-training Data|NWP accuracy (en-US)|NWP accuracy (en-IN) |
> | -----| -------|-----|
> |C4-784G (baseline)|0.1176 | 0.0423 |
> |LLM-filter-C4-136G |0.1415 | 0.0476|
> |LLM-mix-166G | 0.1440 | 0.0506 |
> | LLM-filter-C4-136G  + LLM-converted-C4-10G  (new) |0.1435 |0.0492|
>
> Note:
>
> - For fair comparison, the numbers were collected this week (May 27-29) from 100M+ examples of 200k+ mobile devices. That's why the first 3 rows are slightly different from Table 3 of our paper. Note that this week overlaps with Memorial Day, which may affect the result.
>
> - The new data (last row) is slightly worse than LLM-mix-166G, indicating the potential of only relying on C4. Due to resource limitations, we only converted ~20% of filtered C4 to chat. If given enough resources, it is interesting to convert all C4 and see how it performs.
>
> - Last, the diversity issue that we learned from directly prompting LLMs (vocab coverage in Table 2) is interesting, and worth sharing with the community.

---

> > ### Comment · Reviewer_NTtL · 2024-06-05
> > **Thanks**
> >
> > thank the authors for their reply; I confirm my assessment of the paper.

---

### Official Review · Reviewer_cRaX · 2024-05-10

**Rating:** 5
**Confidence:** 4
**Ethics Flag:** 1

**Summary:**

This paper explores the enhancement of pre-training data quality for on-device language models through the use of large language models (LLMs), specifically within the context of a production mobile keyboard application employing differential privacy and federated learning. The authors propose a new approach where LLMs are utilized to filter and transform public data, as well as to generate new synthetic data, resulting in significant improvements in next-word prediction accuracy (19.0% and 22.8% relative improvement over baseline models trained on standard public datasets). The study demonstrates the LLMs' capability to synthesize data that closely mimics the private distribution, despite not having direct access to private data. Furthermore, the paper identifies potential future improvements to minimize the distribution gap further. This research underscores the transformative potential of LLMs in enhancing data privacy and efficiency in machine learning applications. However, the work somehow can not fully garanttee the privacy protection in a formal way. Moreover, the work also needs to quantify the privacy protection performance.

**Questions To Authors:**

Let LLM to generate data with similar distribution to the private data is somehow empirical which cannot garanttee the privacy protection. Can the authors provide quantitive expeirments for validation? Or provide DP style theoretical garanttee?

**Reasons To Accept:**

1). The work is very interesting.

2). The paper is well presented and easy to follow.

3). The promblem is an important issue.

**Reasons To Reject:**

1), In their current format, the methodologies presented within this work lack a formal guarantee of privacy protection, which might raise concerns about their practical applications in environments with stringent privacy requirements. Additionally, there is an evident need for the work to establish quantifiable metrics that can accurately evaluate the level of privacy protection being afforded. This would not only bolster the trust in the system but also provide clearer insights into the effectiveness of the privacy-preserving features of the synthetic data generation process.

2). Let LLM to generate data with similar distribution to the private data is somehow empirical which cannot garanttee the privacy protection.

---

> ### Author Rebuttal · Authors · 2024-05-31
>
> Thanks for your time and valuable feedback! Below we address your concerns on the privacy guarantees of our method:
>
> In our paper, the private data are user typing data in the on-device mobile keyboard applications. In **Section 3**, we design LLM prompts to synthesize data similar to user tying data without access to any private user data, so the manual prompts are public and carefully examined. As also remarked in **footnote 1** of our paper: "Pre-trained LLMs are considered to be public because their training data do not contain the on-device user data in mobile keyboard applications. The privacy concerns of LLMs and their training data is an important independent topic (Brown et al., 2022; Tramer et al., 2022)."
>
> In **Section 4**, the on-device LMs are pre-trained on public data (either C4 dataset or the synthetic data from LLMs), and then fine-tuned on user data with federated learning (FL) and differential privacy (DP). The device/user-level DP guarantees are detailed in **Appendix C.1**, which are widely used in the literature (McMahan et al. 2018, Kairouz et al. 2021a, Xu et al. 2023). By improving the public pre-training data, we achieve a better privacy-utility trade-off than previous work.
>
> **Section 5** is a preliminary study, where we want to see if DP FL fine-tuned LM can be used to improve the server-side data. Note that the goal here is **not** training new on-device LMs, instead, the goal is to see if we can use an **existing** privately-trained LM to get a better server-side dataset. The obtained dataset could be of independent interest, e.g., used for server-side research simulations. Specifically, the pipeline that we explore in Section 5 is:
>
> - pre-training on LLM-mix-166g + private fine-tuning via FL -> privately-trained LM (used for on-device inference, same as Section 4) -> use the privately-trained LM to filter LLM-mix-166g to obtain LLM-prox-32g.
>
> This pipeline gives two outputs: 1) privately-trained LM (same LM obtained in Section 4), and 2) a LM filtered server-side data. The private user data is **only** used in training the private LM. Obtaining the LM filtered server-side data does not incur **extra** privacy cost, following the post-processing property of differential privacy. Therefore, the pipeline we considered in Section 5 has the **same** DP guarantee as that given in Section 4 and Appendix C.1.

---

> > ### Comment · Reviewer_cRaX · 2024-06-04
> > **Concerns not addressed**
> >
> > Thanks for the response. However, my concerns as in the initial feedback are not well addressed. I would like to keep my score.

---

> > > ### Author Response · Authors · 2024-06-04
> > > **Thanks for the reply, and could you please elaborate?**
> > >
> > > Thanks for your reply. Could you please elaborate on the specific concerns if possible? In the rebuttal above, we have clarified that
> > >
> > > - We adopted the widely acknowledged ($\epsilon, \delta$) differential privacy (DP) notion to quantify the privacy protection when training on real-user on-device data. Our Appendix C.3 not only provided the ($\epsilon, \delta$) values, but also carefully discussed all the nuances for the DP guarantees to hold including data access, unit, neighboring, and accounting methods. Appendix C.3 applies to the DP federated learning training process.
> > >
> > > - We also carefully explained why the LLM synthetic data generated in **Section 3** are treated as public data in our paper, because the private on-device real-user typing data is **not** used during the LLM data synthesis phase.
> > >
> > > - The preliminary study in Section 5 adds a post processing step to the previous pipeline considered in Section 1-4, and has the same DP guarantee (see our rebuttal above).
> > >
> > > We greatly appreciate your comment, and we would hope to get concrete feedback to better address your concerns.

---

### Official Review · Reviewer_gADZ · 2024-05-10

**Rating:** 7
**Confidence:** 3
**Ethics Flag:** 1

**Summary:**

Paper studies the question of how to effectively use public text data to pretrain LMs which are then fiinetuned on private data using DP-FL algorithms. Their application is next work prediction in mobile keyboards. They use an LLM to
1. Filter private data to make their distribution similar to mobile chat data
2. Generate synthetic chat data from scratch for their usecase, and
3. Convert non-chat data (from Step 1) in public dataset to chat like data

Step 1 generated lots of data. Authors identified that the small amount of synthetic data from Step 2 had low vocabulary coverage on private data. So they combined it Step 3 which brought up the coverage.

They tested on two different private datasets. Model on pretrained on Step 1 data was better than model pretrained on unfiltered data. Models pretrained on Step2+3 data sits in the middle in terms of performance. Finally model pretrained on Step1+2+3 was better than all others.

DP-FL finetuning of Step1+2+3 model on the private datasets further improved their performance. Finally they also showed better performance of this model in an online A/B test.

They also studied using privately finetuned LM to futher filter their pretraining dataset. This seems to help in their preliminary study.

**Questions To Authors:**

See above.

**Reasons To Accept:**

1. Paper is easy to read
2. Paper validates the use of LLMs in Public data pretraining+Private data DP-FL finetuning paradigm
3. Application is large scale and authors have done online tests
4. These will be useful for other similar applications.
5. Authors identified some important future research directions

**Reasons To Reject:**

1. It is not clear if authors have done enough literature survey. I found this paper from last year which is very related. “Privately customizing prefinetuning to better match user data in federated learning” https://arxiv.org/pdf/2302.09042. Please discuss this paper and any other idea from literature.
2. Not to surprising. But I would not want to use this as a main reason for recommending rejection.
3. Last section on using privately finetuned LM to filter public data reads a bit like a last minute add on. It is not well motivated and the results are only partial. It is not clear how privacy is accounted here. Please expand on these.

---

> ### Author Rebuttal · Authors · 2024-05-31
>
> Thanks for your time and valuable feedback!
>
> > 1. "...I found this paper...Please discuss this paper..."
>
> Thanks for suggesting this paper. Below we discuss the differences (will add them to our paper):
> - Hou et al. (2023) proposed FreD, a method to measure the distance b/t a server-side dataset and the private federated dataset on users’ devices. FreD metric is used for server-side **dataset selection**.
> - By contrast, we use LLMs to **synthesize** good server-side datasets.
> - FreD can be combined with our approach, e.g., use LLMs to get a few datasets (our paper), and use FreD to select a good one. This requires careful allocation of privacy budgets in FreD and federated fine-tuning.
>
> We have discussed a few related work in our paper, e.g., **Wang et al., 2023** (see Introduction) and **Zhang et al., 2023** (see Section 2).
>
> This is an active research area, and a few new papers appear **after** our COLM submission, e.g.,
> - Hou et al., https://openreview.net/forum?id=SbyqM0cJM0, 2024
> - Li et al, https://arxiv.org/abs/2405.14212, 2024
>
> Compared to the new papers, our work provides unique value by running real-world federated learning (FL) experiments.
>
> > 2. "Not too surprising..."
>
> We agree most of the results match the expectations, and thanks for clarifying that this is not the main factor. We highlight that this is the first time the results are observed in a production FL system: e.g., LLM synthetic data are closer to the private distribution but there is still a gap (Figure 4); LLM filtered C4 is better than LLM generated chat (Table 3); using a privately-trained LM to filter data only gives a slight gain (Table 4). We hope that our work can inspire more future research.
>
> > 3. "Last section on using privately finetuned LM to filter public data......"
>
> Section 5 presents a preliminary study on whether a privately-trained LM can help reduce the distribution gap between the server-side data and the private user data. The goal here is **not** training new on-device LMs, instead, the goal is to see if we can use an **existing** privately-trained LM to obtain a better dataset. The obtained dataset could be of independent interest, e.g., used for server-side research simulations. The private user data is **only** used in training the on-device LM (so it has the same privacy guarantee as before, see Appendix C.1). Using this LM to filter server-side data does not incur **extra** privacy cost, following the post-processing property of differential privacy.

---

> > ### Comment · Reviewer_gADZ · 2024-06-05
> > **Reply**
> >
> > Thanks for the response to my concerns. Authors addressed most of my concerns, so I will raise my score while requesting the authors to incorporate this discussion to their manuscript.

---

> > > ### Author Response · Authors · 2024-06-05
> > > **Thanks!**
> > >
> > > Thanks! We will incorporate these discussions to our paper.

---

### Decision · Program_Chairs · 2024-07-10

**Decision:**

Accept

**Comment:**

In this paper, the authors design LLM prompts to filter and transform existing public data, generating new data that closely resembles real user data distributions. The model pre-trained on this synthetic dataset achieves better performance compared to the baseline model pre-trained on a standard public dataset. Furthermore, the model pre-trained by the synthetic data demonstrates comparable or superior performance to the baseline during differential private federated fine-tuning on user data from millions of mobile devices and outperforms the baseline in production A/B testing. Following the author response and subsequent discussions with reviewers, the paper has obtained sufficient support from the majority of reviewers. Therefore, I recommend acceptance.